# Long-Term Exposure to Lambda-Cyhalothrin Reveals Novel Genes Potentially Involved in *Aedes aegypti* Insecticide Resistance

**DOI:** 10.3390/insects16020106

**Published:** 2025-01-21

**Authors:** Alejandro Mejía, Ana María Mejía-Jaramillo, Geysson Javier Fernandez, Yurany Granada, Carl Lowenberger, Omar Triana-Chávez

**Affiliations:** 1Group Biología y Control de Enfermedades Infecciosas, Universidad de Antioquia UdeA, Calle 70 No. 52-21, Medellín CP 050010, Colombia; alejandro.mejiam1@udea.edu.co (A.M.); maria.mejia3@udea.edu.co (A.M.M.-J.); geysson.fernandez@udea.edu.co (G.J.F.); yurany.granada@udea.edu.co (Y.G.); 2Centre for Cell Biology, Development, and Disease, Department of Biological Sciences, Simon Fraser University, 8888 University Drive, Burnaby, BC V5A 1S6, Canada; clowenbe@sfu.ca

**Keywords:** transcriptome, *Aedes aegypti*, insecticide resistance, lambda-cyhalothrin

## Abstract

*Aedes aegypti* is a mosquito vector of viruses such as dengue, Zika, and yellow fever, which affect millions of people around the world. Chemical insecticides are the primary measure health authorities take to control this mosquito; however, continuous exposure increases the resistance to these insecticides. In this study, we evaluated the response of this mosquito to long-term exposure to lambda-cyhalothrin and compared the differential gene expression between exposed and non-exposed mosquitoes. This study presents a novel perspective on the response of *Aedes aegypti* to lambda-cyhalothrin insecticide. After long-term exposure, mosquitoes respond to insecticides by expressing different proteins. Overall, we propose a model that incorporates previously unexplored mechanisms found in response to prolonged insecticide exposure, along with the established and accepted insecticide resistance mechanisms such as *kdr* and metabolic resistance.

## 1. Introduction

The mosquito *Aedes aegypti* transmits arboviruses, including dengue, Zika, and chikungunya, which impact global health. The diseases produced by these viruses cause nearly 400 million infections and 100,000 symptomatic cases, resulting in 40,000 deaths annually [1]. The Americas witnessed a concerning rise in arboviral prevalence, driven by climate change-induced meteorological phenomena, perpetuating epidemic cycles [2].

Insecticides used as vector control persist as the primary form of intervention. Chemicals such as organophosphates, carbamates, insect growth regulators (IGRs), and pyrethroids control adult and larval forms. Lambda-cyhalothrin, a type-II pyrethroid, is Colombia’s most commonly used insecticide [3]. However, decades of extensive application have exerted high selective pressures on wild populations of *Ae. Aegypti,* significantly hampering vector control efforts by increasing their resistance to insecticides [4]. Lambda-cyhalothrin resistance is present in at least 76% of *Ae. aegypti* populations in Colombia, which is concerning due to their distribution [3,5,6,7,8,9].

Many studies have focused on the classical mechanisms [10], knockdown resistance (*kdr)* mutations, and metabolic enzymes (also called metabolic resistance) to understand the development of insecticide resistance. *Kdr* mutations are modifications in the Voltage-Gated Sodium Channel (VGSC) gene-encoding region that impair the insecticide’s ability to bind to its molecular target. For example, the mutations F1534C and V1016I have been found in the VGSC gene of resistant field mosquitoes and are correlated with elevation in LC_50_ (Lethal Concentration 50) to pyrethroid insecticides [11]. Other studies also found the contribution of these alleles to lambda-cyhalothrin insecticide resistance [12], including the mutation V419L [8,13]. Likewise, metabolic resistance entails an increase in the activity of detoxifying enzymes such as those that belong to the Cytochrome P450 (CYP450) family, glutathione S transferases (GST), and carboxylesterases (CE) [14].

The artificial selection of *Aedes aegypti* with insecticides allows the identification of features that could not be found in wild populations. Characterizing the molecular mechanisms is crucial for vector control within this framework, and tracking insecticide-resistance genes is vital for management strategies. Recently, some *Ae. aegypti* strains were selected with pyrethroid insecticides over a few generations, and an RNA-seq approach was used to find genes associated with resistant populations [15]. However, little is known about the response to long-term exposure to the lambda-cyhalothrin insecticide over several generations in *Ae. aegypti*.

In this study, we subjected a strain of *Ae. aegypti* to selective pressure for 13 consecutive generations to understand the development and extent of insecticide resistance. We assessed resistance ratios and the magnitude of resistance in these mosquito populations. Then, *kdr* mutation typing and enzymatic activity assays were carried out to identify whether the known resistance mechanisms were present in this resistant mosquito population. Lastly, we delved into the transcriptomics of this pressured strain, comparing it to that of the same strain without pressure to gain insights into the molecular changes underlying insecticide resistance in *Ae. aegypti*.

## 2. Materials and Methods

### 2.1. Study Area and Mosquito Collections

The *Ae. aegypti* population used in this study was collected in 2016 in Acacías, Meta, Colombia (3°59′20″ N 73°45′53″ W) [16]. The mosquitoes were maintained under controlled conditions with a temperature of 25 ± 5 °C, relative humidity of 72% ± 5%, and a photoperiod of 12 h light/12 h dark. Larvae were raised in dechlorinated water and fed Purina^®^ truchina fish food (48% protein). Upon reaching adulthood, they were transferred to cages measuring 50 × 50 × 50 cm and provided with 10% sugar solution *ad libitum*. The females were offered a blood meal from a mouse. After blood feeding, a strip of moist filter paper was immediately placed in each cage as an oviposition substrate. The egg strips were removed and stored in humidity-controlled boxes for future use.

### 2.2. Experimental Design

To study the effects of long-term exposure to insecticides on *Ae. aegypti* populations, the parental Acacías population named AF-0 was used. This population was previously reported to have a resistance ratio (RR) of 31.4 to lambda-cyhalothrin [16]. AF-0 was subsequently separated into two batches. The first batch was pressured with lambda-cyhalothrin at a concentration that generally would kill 90% of the insects (LC_90_) for 13 generations and was given the name AFP (Acacías pressured). The second was an independent batch of Acacías AF-0 reared under standard laboratory conditions for 13 generations without insecticide pressure, named AFWP (Acacías without pressure). For each batch and each generation, we used at least 1000 individuals. Bioassays were performed on AFP and AFWP generation 13, also named AF13P and AF13WP, respectively (Figure 1). *Kdr* typing, enzyme metabolic activity, and RNA sequencing (RNA-seq) were performed on both populations. Additionally, mosquitoes from generation F7 of Acacías pressured (AF7P) and unpressured (AF7WP), previously reported by Granada et al. [16], were used to compare RR and allelic frequencies.

### 2.3. Larval Bioassays and Pressure

Larval bioassays were used to obtain the LC_90_ of AF-0. For this, WHO protocol [17] with technical grade lambda-cyhalothrin (99.8% a.i.; CAS number 91465-08-6) from Sigma-Aldrich (Burlington, MA, USA) was used. Batches of 20 larvae (with three biological replicates) of the third and fourth instars were introduced into 99 mL of distilled water and one mL of insecticide resuspended in ethanol, and mortality was recorded after 24 h. One percent ethanol was used as a control in the bioassays. Insecticide concentrations were 0.0009 to 0.06 ppm and were based on mortality ranges from 10% to 90%. The results obtained in the bioassays were subjected to probit analysis using the SPSS toolkit (IBM SPSS Statistics for Windows, Version 27.0) to determine the LC_50_ and LC_90_. The LC_90_ was used for insecticide pressure in each mosquito generation. Once generation 13th was reached, the bioassays with AF13P and AF13WP were carried out to obtain the corresponding RR. The susceptible population Rockefeller was used as a reference. The RR was calculated by dividing the LC_50_ (or LC_90_) values of the population under study by that obtained for the susceptible strain. This calculation yields a 50 (RR_50_) or 90 (RR_90_) resistance ratio. The statistical significance (*p* < 0.05) of the comparison of LC_50_ and LC_90_ between resistant and susceptible strains was assessed using the Lethal Dose Ratios test [18]. This method involves the overlap of the 95% confidence intervals (95% CI) for each evaluated strain compared with the 95% CI of the Rockefeller strain. If there is no overlap between CI, it is a significant difference with *p* < 0.05. After accomplishing this procedure, AF13P and AF13WP were used to determine *kdr* mutations and enzymatic activity.

### 2.4. Determination of Kdr Allelic Frequencies

An allele-specific PCR (AS-PCR) within the voltage-gated sodium channel gene-encoding region was used to amplify the known *kdr* mutations F1534C, V410L, and V1016I circulating in Colombia [8,13]. Genomic DNA was extracted using the Grind Buffer protocol [19]. Each insect was homogenized in 50 µL of buffer and then incubated with 20 µL of proteinase K. After adding potassium acetate [8M], centrifuge steps were performed to precipitate DNA, followed by washing with 70% and 98% ethanol. The resulting DNA pellet was resuspended in 30 µL of water. Subsequently, each PCR reaction was performed in a Rotor-Gene Q thermocycler, following conditions reported by Pareja-Loaiza et al. [13]. At least 30 adult mosquitoes from the insecticide-pressured and unpressured populations were used. Additionally, wild-type alleles from the Rockefeller population were used as a reference.

### 2.5. Metabolic Enzyme Activity

Enzyme activity assays were performed to identify metabolic resistance in insecticide-pressured (AF13P), unpressured populations (AF13WP), and the Rockefeller reference strain. The enzymes assessed included acetylcholinesterase (AChE), alpha and beta esterases (α-EST and β-EST, respectively), mixed-function oxidases (MFO), and glutathione S-transferase (GST). Forty female mosquitoes that had emerged within the last day were individually macerated in 300 µL of deionized water using a tissue grinder and a MicroPestle system until the sample was homogenized entirely. For the metabolic enzymatic activity assays, we followed standardized protocols [20]. The enzymatic activity of acetylcholinesterase (AChE), mixed-function oxidase (MFO), α-esterase (α-EST), β-esterase (β-EST), and glutathione-S-transferase (GST) was determined. All enzyme activity analyses were normalized with the total protein concentration, determined using the Pierce BCA Protein Assay Kit (Thermo Scientific, Rock, Waltham, MA, USA), with 10–25 µL of mosquito homogenate or supernatant following the manufacturer’s instructions. Briefly, for the AChE activity, 25 µL of homogenized mosquito in the presence and absence of inhibitor was determined by adding DTNB (5,5-dithio-bis-(2-nitrobenzoic acid)) substrate. For MFO, 20 µL was used, and the substrate was TMBZ (3,3′,5,5′-Tetramethylbenzidine). For α-β-EST, 10 µL of supernatant was pipetted, and α-β naphthyl was used as a substrate. Finally, reduced glutathione as a substrate and 15 µL of supernatant were used for GST. All enzymes were measured in an ELISA Multiskan Spectrum from Thermo Fisher Scientific using wavelengths previously reported [21].

### 2.6. RNA Sequencing and Bioinformatic Analyses

For the 2 populations, AF13P and AF13WP, three pools of five female mosquitoes were employed for each RNA extraction using the Spin Tissue RNA Mini Kit. Briefly, mosquito pools were homogenized using micropestles and lysed using a lysis buffer. RNA was eluted in 30 µL of TE buffer. The samples were sent to the University of Oklahoma (Norman, OK, USA) for sequencing. For library preparation, mRNA was purified using poly-A tails, and the Illumina NovaSeq 6000 kit (Illumina, San Diego, CA, USA) was used to obtain paired-end reads of 150 base pairs. The average sequencing depth was 20 million reads. Sequence quality was verified using FastQC [22], and low-quality sequences were cut using Trimmomatic (Phred Score < 30, length < 36) [23]. The AaegL5 genome [24] (Accession: GCA_002204515) was used as a reference for mapping using STAR [25]. This genome contains 19,804 genes, with 14,718 coding for proteins. Gene counts were obtained through the Rsubread (DEseq2) package, and the Differentially Expressed Genes (DEGs) were defined based on an absolute log 2-fold change (log_2_FC) ≥ 1.0 and a False Discovery Rate (FDR) ≤ 0.05 between the two compared groups (AF13P vs. AF13WP). Normalized data were scaled for statistical analysis and subsequent comparisons between the two groups.

### 2.7. Functional and Enrichment Analysis

The DAVID Gene system was used to elucidate resistance mechanisms by associating overrepresented genes in the RNA-seq data with specific biological processes in the Gene Ontology (GO) database and performing pathway enrichment analysis using KEGG [26]. The functional analysis and classification were separated into upregulated and downregulated terms using all expressed genes as a background. DAVID uses the EASE Score for functional analysis; it is a modified Fisher exact *p*-value to test the probability of getting a gene (or a set of genes) from our whole data set, whose associated term is obtained with a particular frequency, over the total gene set of a background genome (in this case, *Aedes aegypti* AegL5 genome) and asks if there is more than randomness in selecting that specific term. Gene clustering was made with the option “Functional Annotation Clustering,” using a kappa statistic to prevent repeated annotations from being overrepresented in the gene list. Additionally, the enrichment score is used to classify the grouped genes, and the greater the score, the more significant the classification. An enrichment score ≥ 1.3 equals a *p*-value of 0.05 (10^−1.3^).

In summary, the enrichment score groups terms with similar biological meanings by having similar gene members. It is based on the EASE score, a modified statistical test. The smaller this score is, the more enriched the term (and therefore the gene) is in the list.

The functional classification was performed by grouping genes with a similar function to improve the biological interpretation of gene lists. The GO annotations were filtered based on the classification performed by DAVID. Moreover, a search was conducted using these terms based on the Molecular Function (MF), Cellular Compartment (CC), and Biological Processes (BP) categories. Functional annotation, clustering, and GO analysis were plotted using SRplot “https://www.bioinformatics.com.cn/en (accessed on 10 July 2024)”, a free online data analysis and visualization platform.

### 2.8. Ethics Statement

All animals used in this study were handled strictly according to good animal practice, as defined by the Colombian Code of Practice for the Care and Use of Animals for Scientific Purposes, established by Law 84 of 1989. Ethical approval (Act No. 136, 17 November 2020) was obtained from the Animal Ethics Committee of the University of Antioquia, Medellin, Colombia.

## 3. Results

### 3.1. Prolonged Exposure to Lambda-Cyhalothrin Induces a High Degree of Resistance

We explored the effect of insecticide pressure on the RR of AF13P compared to AF13WP using bioassays. The unpressured population exhibited a reduction in RR50 of approximately 1.6 times each generation. While the AF-0 strain showed an RR50 of 31.64, the AF7WP and AF13WP strains had significantly lower values of 18.99 and 10.54, respectively, indicating greater susceptibility after a long time without insecticide pressure. Conversely, after the insecticide pressure, the AF13P population increased the resistance, reaching an RR50 of 748.94, 23 times higher than the parental AF-0 strain and 71 times higher than the AF13WP. The RR50 increased drastically after seven generations under insecticide pressure (AF7P, RR50 662.45) (Table 1). Nonoverlapping, 95% CIs were observed between evaluated strains, indicating a significant difference (*p* < 0.05) between all Acacías strains and Rockefeller.

Our selective pressure experiment yielded a highly resistant strain (AF13P) compared to parental AF-0 and even more resistant than the unpressured AF13WP, showcasing the selection or emergence of resistance mechanisms, which we next explored.

### 3.2. Low Kdr Allelic Frequencies Do Not Rule out High Lambda-Cyhalothrin Resistance in Aedes aegypti

Since the presence of specific *kdr* mutations, such as V410L, V1016I, and F1534C, are known to be involved in insecticide resistance over successive generations, we sought to identify the frequencies of these mutations in 30 individuals of AF13P and AF13WP and compare them to those of AF-0, AF7P, and AF7WP previously reported [16]. The mutated allele 410L frequency increased from 0.4 to 0.6 in AF7WP and AF13WP, respectively (Figure 2A). A drastic change occurred between pressured strains AF7P and AF13P, with mutated allele frequency dropping from 0.58 to 0.02, respectively. The same tendency was observed in the 1016I mutation for both the unpressured and pressured populations. The 1016I allele in AF7WP and AF13WP mosquitoes experienced an increase in frequency from 0.33 to 0.46, respectively. Conversely, AF7P and AF13P individuals diminished their mutated allele frequency from 0.58 to 0.05 (Figure 2B). Nonetheless, the allelic frequency of F1534C remained constant in susceptible and resistant populations, being fixed in tested populations (Figure 2C).

### 3.3. Metabolic Enzyme Activity Is Not Showing Significant Changes in Acacías Populations

Given the unexpected findings in the analysis of *kdr* mutations, we looked for alternative mechanisms involved in resistance, such as metabolic resistance. Thus, enzymatic activities were determined in AF13WP and AF13P (Figure 3). It was apparent that even after prolonged exposure to lambda-cyhalothrin over 13 generations, the AF13P exhibited no substantial increases in enzymatic activity across most enzymes compared to AF13WP, specifically α-EST, β-EST, and GST. Interestingly, AF13P exhibited a slight increase in ACHE activity compared to AF13WP, while MFO levels were increased in AF13WP. However, it is to be noted that both Acacías strains had low ACHE and MFO and high α-EST and GST when compared with the susceptible reference strain Rockefeller. These results suggest that some enzymes presented altered levels, potentially indicating adaptive mechanisms or resistance maintenance in these populations when compared with Rockefeller. Nonetheless, the AF13P-resistant population did not portray representative changes in enzymatic activity compared to its unpressured counterpart AF13WP, from which we could not account for the high RR previously obtained (Table 1).

### 3.4. Identifying Potential New Insecticide Resistance Mechanisms Through RNA-seq

Our data showed that mosquitoes highly resistant to lambda-cyhalothrin do not have a high frequency of mutated alleles or altered activity of selected metabolic enzymes, so additional resistant mechanisms should be involved. Therefore, we employed RNA sequencing to identify genes differentially expressed in the insecticide-resistant strains that might contribute to understanding the insecticide resistance phenotype.

We obtained 790,369,834 total trimmed reads, with approximately 93% mapping coverage to the reference AegL5 genome (Appendix A). Trimmed reads were used for mapping using STAR with the latest *Ae. aegypti* genome version (LVP_AGWG AaegL5.3). After filtering low-expressed genes (<32 gene counts), 12,893 (65.1%) genes remained, of which 78.79% were uniquely mapped genes and 13.64% were mapped to multiple loci.

DEseq2 was used to identify differentially expressed genes (DEGs) between AF13P and AF13WP populations. Of the initial 12,893 genes, 320 were upregulated, and 602 were downregulated in the AF13P-resistant population compared to AF13WP-susceptible (Figure 4A). Normalized counts of DEGs were used to construct a PCA, revealing a distinct separation in expression data between AF13P and AF13WP populations for the Principal Component 1 (PC1) and 2 (PC2), contributing to a cumulative explained variance of 70.3% (Figure 4B). Among the 320 upregulated genes, 231 were protein-coding genes: 150 corresponded to unspecified products (hypothetical genes), and 81 were annotated (Appendix A). Of the 602 downregulated genes, 475 were protein-coding genes. Among them, 230 were successfully annotated, while 245 were reported as hypothetical genes (Appendix A).

To identify genes associated with insecticide resistance, we explored the genes with the highest and lowest FC (Fold Change). The most upregulated genes were AAEL019996, AAEL025347, AAEL025157, and AAEL023405, which are not annotated in the genome.

Gene AAEL019996 is 8.9 times upregulated and belongs to a probable 34 kDa protein reported as a salivary-secreted peptide in Uniprot [27]. The upregulated genes AAEL025347 (FC = 10.1, FDR = 7.08 × 10^−13^), AAEL025157 (FC = 8.20380676, FDR = 6.06 × 10^−8^), and AAEL023405 (FC = 7.969957451, FDR = 6.96 × 10^−6^) correspond to ncRNAs (non-coding RNA) (Appendix A, DEGs).

The top 3 downregulated genes, AAEL018681, AAEL018658, and AAEL018671, code for subunits of NADH dehydrogenase (FC = −13.4736225, −13.1435047, and −11.2037335, respectively). Additionally, gene AAEL018661 ranks the least expressed in all datasets, with a log_2_FC of −14.1, corresponding to tRNA-Tyr (Transfer RNA-Tyrosine).

### 3.5. Prolonged Exposure to Lambda-Cyhalothrin Reveals Clusters of Genes from the Cuticle and Respiratory Chain

After identifying DEGs in the resistant strain, we studied the biological role of upregulated and downregulated genes and analyzed their relationship with insecticide resistance. For this purpose, we used DAVID for functional annotation, clustering (Appendix A), and Gene Ontology (GO) to analyze gene functions (Figure 5).

Using DAVID on the list of 320 upregulated DEGs within the broader context of the entire dataset, nine distinct clusters were revealed (Appendix A). Only two were significant, displaying an enrichment score above 1%. The first gene cluster, characterized by an enrichment score of 4.3%, includes a conserved protein domain SCP (Sperm-Coating Glycoprotein) and pathogenesis-stress-related protein, P14a type (Appendix A). The second cluster (enrichment score of 2.5%) comprises terms directed toward cuticle structure. Finally, cytochrome P450, although having an enrichment score < 1, emerged from the data as a third cluster (Appendix A).

According to GO term separation, the overexpressed genes with the most significant *p*-value are those of structural constituents of the cuticle (GO:0042302) and odorant binding (GO:0005549) (Figure 5A), portraying a similar picture between DAVID clustering and GO.

The 602 downregulated genes included six statistically significant clusters (enrichment score above 1). Clusters one and two were dominated by enzymes, mainly trypsin, peptidases, proteases, collagenases, and transmembrane proteins. The third cluster comprised genes related to cytochrome P450, iron, heme binding, and respiratory chain enzymes (Appendix A); the terms GO:0005506 ion binding and GO:0004129 cytochrome-c oxidase activity were also significant. The fourth and fifth clusters included terms related to transmembrane components and transmembrane activity, such as GO:002857 transmembrane transporter activity (Figure 5B). The last cluster, showing an enrichment score above 1, incorporated leucine-rich repeats, but GO terms did not represent it.

In summary, DAVID allowed the identification of genes into organized categories. GO corroborated this analysis by portraying overlapping GO categories and enrichment gene terms. In terms of insecticide resistance-related genes, the most exciting categories for the reported DEGs were cuticle protein genes for upregulated genes and chain transporter and stress-related proteins (proteases, peptidases, and leucine-rich repeat genes) for the downregulated genes. We included four additional categories of metabolic resistance genes to expand our analysis further and provide context for the high Resistance Ratio (RR) observed in Table 1 and the varying degrees of enzymatic activity in our strains (Figure 3). These categories comprised GST, esterase, dehydrogenase, and CYP450. We examined the FC distribution of these genes to gain a more comprehensive understanding of their potential role in insecticide resistance mechanisms.

### 3.6. Prolonged Exposure to Lambda-Cyhalothrin Reveals Broad Distribution of DEGs Among Gene Categories

According to DAVID’s classification, we displayed the log_2_FC of each gene category that was significant (*p*-value < 0.05) in Figure 6. This organization compares the FC of genes with the same biological function and explains their potential role in insecticide resistance based on the relationship between upregulated and downregulated genes. Additional information on each of the displayed genes, log2FC, and VectorBase annotation is shown in Appendix A.

Accounting for all the genes in the categories, we showed that only 17 of the 116 total genes were upregulated (Figure 6), exhibiting a similar tendency of low quantity of upregulated vs. downregulated DEGs (320 vs. 602 genes). Of the total 17 genes, eleven upregulated genes belonged to metabolic enzyme categories. The six remaining upregulated genes corresponded to the cuticle category (AAEL007739- pupal cuticle protein, AAEL011504- pupal cuticle protein, AAEL013380- adult cuticle protein). It is to be noted that enzymatic-related categories, CYP450, dehydrogenase, GST, and esterase, showed a bimodal distribution of log_2_FC.

The CYP450 category depicts three upregulated genes. The genes encoding for CYP9J29, CYP6AG4, and the CYP450- AAEL009018 of an unknown family (Appendix A). Of the 15 downregulated genes, the majority of them correspond to the family of CYP6 (8 genes), followed by the family CYP4 (3 genes) and CYP9 (2 genes), with only 1 gene of the CYP12 family. The most downregulated gene of this category is CYP6F3, which has an FC of −4.03.

In the dehydrogenase category, ten genes are significant, and of those, AAEL013603 (log_2_FC = 2.8) is one of the most upregulated genes that encodes a short-chain dehydrogenase (Figure 6). In contrast, the downregulated gene AAEL011130 is identified as alcohol dehydrogenase, exhibiting a log_2_FC of −2.1.

These bimodal distributions exhibit genes with distinct responses to lambda-cyhalothrin pressure according to their log_2_FC, suggesting varied sensitivities among these enzymes. We showed that Glutathione transferase genes GSTX2 and GSTX1 were upregulated, whereas the snRNA phosphodiesterase AAEL009090 was the only gene of the esterase category with elevated transcription. Interestingly, two GSTs and microsomal glutathione s-transferase were downregulated, while the only downregulated gene of CCE is CCEAE1A (Appendix A and Figure 6).

Surprisingly, electron chain transporter genes had downregulated genes with the lowest value compared to other categories. They corresponded to NADH dehydrogenases and cytochrome c oxidases with a log_2_FC ranging from −1.2 to −13.4 (Figure 6 and Appendix A).

On the other hand, the protease group had the most downregulated genes, with 56 genes involved in protein production, of which only three were significantly upregulated (AAEL007420—serine protease, AAEL012219—ubiquitin-specific protease, AAEL014946—protease U48). Another group separated by DAVID was the peptidases and LRR (Leucine-Rich Repeats), with six and five significant downregulated genes each. Serine type endopeptidase and LRIM1 are the least downregulated exponents of peptidases and LRR categories. It is to be noted that the most downregulated gene in our data set (AAEL018661), which encodes a tRNA-Tyr, does not correspond to any of these classifications and displays a substantial log_2_FC of −14.1 (Figure 5).

## 4. Discussion

This study measured the differential expression profile in an *Ae. aegypti* strain that exhibits a significant increase in resistance to lambda-cyhalothrin, independent of *kdr* mutations and metabolic resistance mechanisms. We found several genes that could be responsible for heightened resistance.

### 4.1. Classical Mechanisms of Insecticide Resistance Are Underrepresented in AF13P

Mosquitoes typically exhibit primary resistance mechanisms through VGSC gene alterations and metabolic enzymes [28,29,30].

The frequencies of the mutated alleles in the highly insecticide-resistant AF7P strain were significantly higher compared to the AF13P strain. Specifically, the frequency of the 410L mutated allele decreased from 58% to 5%, while the frequency of the 1016I mutated allele dropped from 58% to 2%. Similarly, other studies have found that the *kdr* frequencies may decline among resistant populations due to fitness costs [31,32,33]. However, these findings differ from previous studies that examined the sequential evolution of the 1534C and 1016I mutations [34], where co-occurrence of these alleles was associated with increased resistance to pyrethroids. Additionally, it was suggested that the combination of 410L + 1016I + 1534C alleles results in high fitness costs without insecticides [33], but in the presence of deltamethrin, this combination enhances resistance compared to having only the 1534C allele [35]. Other *kdr* mutations have been reported [4,28] that were not identified by our molecular approach; however, we screened for the most associated mutations in the Americas.

Our findings indicate that prolonged exposure to lambda-cyhalothrin across multiple generations maintained the 1534C allele fixed in the population, which could play a key role in insecticide resistance. Since this mutation was fixed in the F7 generation, but the resistance ratio continued increasing, other mechanisms should be involved in expressing insecticide resistance.

Detoxifying enzymes like CYP450 initiate detoxification as a defense mechanism against insecticides [36]. Surprisingly, in metabolic enzymatic assays, we did not observe elevated levels of CYP450 in the resistant AF13P strain compared with the unpressured strain AF13WP, despite their actual differences with Rockefeller. This is coherent with findings from studies in Colombia [4,16,37,38] and elsewhere [39,40], which have suggested a link between this enzyme and resistance to all pyrethroids. Heightened GST enzyme activity has been implicated in varying degrees of resistance to all main classes of insecticides [41]. We did not find significant differences in GST enzyme activity between AF13WP and AF13P mosquito populations (Figure 3). This indicates that this enzyme is unlikely to be a factor responsible for the differences between the resistant AF13P and susceptible phenotype AF13WP (Table 1).

While we do not have sufficient information that explains the differences in resistance between AF13P and AF13WP, our evidence suggests that the discrepancy between these strains and Rockefeller is answered by *kdr* frequencies and enzymatic activity, showcased by RR50 and RR90 in Table 1. Nonetheless, our data suggest that different mechanisms beyond classical ones influence AF13P mosquito resistance.

### 4.2. Diverse Regulation of CYP and GST Genes After Lambda-Cyhalothrin Exposure

Our comprehensive RNAseq analysis provided significant insights into the additional mechanisms related to insecticide resistance in AF13P mosquitoes. Our findings have underscored the crucial role of specific genes in insecticide resistance. Firstly, we detected the upregulation of the CYP6CB1 gene (AAEL009018). This gene was also detected in a modified *Ae*. *aegypti* strain [42], and it is relevant in deltamethrin-resistant field and laboratory *Ae*. *aegypti* populations [43] and within Asian strains [44]. Second, the CYP9J29 gene (AAEL014610), the most over-regulated gene of the CYP450 family in our data set, is a significant locus linked to pyrethroid resistance in *Ae*. *aegypti* [45]. Third, the upregulated CYP6AG4 (AAEL007010) gene was also found with a higher frequency in resistant mosquitoes [46], which probably contributes to insecticide resistance.

In addition, the GST genes GSTX1 (AAEL000092) and GSTX2 (AAEL010500) were also upregulated in resistant mosquitoes (Appendix A). These genes also presented upregulation in Mexican *Ae. aegypti* strains after insecticide selection, aligning with findings from prior studies [47]. Interestingly, these genes do not show upregulation in response to a single instance of pressure generation; instead, they are observed only after four to five generations of sustained pressure.

Transcriptomic studies on lambda-cyhalothrin resistance have only been performed on field-collected mosquitoes that have survived insecticide applications for a few generations [48]. Our insecticide-pressure procedure entails some aspects that should be highlighted. As stated [47], continued exposure to an insecticide portrays the upregulation of new genes that were absent before.

Additionally, a comparative transcriptomic analysis of *Ae. aegypti* explored gene regulation across various time points following exposure to permethrin and revealed that after 24 h of insecticide exposure, there was a notable shift in the number of upregulated (371) and downregulated genes (476) [49]. This suggests a temporal dynamic in the molecular response to insecticide exposure that has to be encompassed, with potential implications for understanding the insecticide’s impact on gene regulation over time. In this sense, it is plausible that the genes identified in our study are associated with long-term exposure responses to insecticide, aligning with previous research findings [47,49]. The transcriptional response to insecticides in resistant mosquitoes involves a complex pattern of upregulation and downregulation of detoxification genes (Figure 6). This suggests that the net production of transcripts, and therefore the overall enzymatic activity detected by biochemical assays, remains relatively stable despite these opposing changes. For example, a study of multiple resistant strains of *Ae. aegypti* reported that CYPs are involved in insecticide resistance; however, their MFO activity was not different between contrasting phenotypes [39]. This emphasizes that solely assessing enzymatic activity may not consistently identify enzymes as responsible for insecticide resistance, challenging the conventional belief associating heightened enzyme activity with resistance mechanisms. Therefore, these studies should be conducted with functional genomics approaches to demonstrate the natural role of new genes in resistance.

We hypothesize that the AF13WP strain may represent an early-stage mechanism of resistance that hasn’t lost its enzymatic activity. Furthermore, with long-term exposure, these enzymes and mutations become less relevant and more costly, opting for a more energy-efficient state, as shown by the AF13P strain. It retains specific genes that could contribute to resistance. These transcripts are further enhanced and preserved in the AF13P strain.

### 4.3. Highly Enriched Constituents of the Cuticle in Response to Prolonged Exposure to Lambda-Cyhalothrin

It is known that the cuticle is the primary barrier that insecticides must overcome to anchor to their molecular targets physically. The enlargement or modification of structural components of this barrier significantly increases the time required to act upon VGSC protein, making it difficult to achieve its intended effect [50,51]. Our findings suggest this mechanism may play a role in the lambda-cyhalothrin highly resistant AF13P strain. Thus, it is not surprising that we found upregulation of cuticle genes (Figure 6) and enrichment of terms related to “structural constituent of the cuticle” GO:0042302 (Figure 5). Genes AAEL007739 and AAEL011504 code for pupal structural constituents, while AAEL013380 codes for adult structural constituents (Figure 6 and Appendix A). According to the literature, these genes have mixed responses and implications. AAEL011504 was significantly upregulated in mosquitoes infected with Zika virus and Wolbachia [52]. AAEL013380 putative adult cuticle protein has an ortholog in *Anopheles albimanus* (AALB005312) from the family of cuticular proteins RR-1 [53]. It has also been identified as downregulated in laboratory-selected mosquito strains with *Bacillus thuringiensis israelensis* toxins [54]. This role of cuticle genes and their mixed cocktail of responses has to be elucidated in more detail, as modifications in these proteins could modify the interconnection between hard, intersegmental, and soft cuticles, providing a complicated environment for insecticide passage [55].

Cuticle modifications are not thought to be a primary mechanism of insecticide resistance, as they are insufficient to elevate the resistance [55]. However, experiments are needed to confirm the relative contribution of this resistance mechanism to overall insecticide resistance [51]. One approach could be to overregulate cuticle protein genes within the background of a susceptible strain such as Rockefeller and test whether RR is modified.

### 4.4. Resistant Mosquitos Exhibit Downregulation of Proteases and Transport Electron Chain Subunits

Proteases are a group of enzymes with multiple functions across insects, including development and digestion. They act by hydrolyzing peptide bonds to separate amino acids and facilitate digestion [56]. Some insecticides act by inhibiting midgut proteases that are fundamental for feeding processes [57]. The insecticide lambda-cyhalothrin has shown different effects on proteases. In *Periplaneta americana,* all cytoplasmic proteases were increased after treatment with this insecticide, but levels of lysosomal proteases in the head and thorax [58] were reduced. In a more related dipteran, *Culex pipiens* protease levels fall after lambda-cyhalothrin treatment [59]. Our study mainly showed downregulation among protease genes (Figure 6), which may contribute to developing a resistant phenotype that offers some degree of protection against insecticides and the oxidative damage resulting from their effects.

On the other side, oxidative stress is caused by an imbalance between producing reactive oxygen species (ROS) and a system’s capacity to detoxify these species. Many insecticides, including lambda-cyhalothrin, induce oxidative stress by increasing the accumulation of ROS. Studies on *Myzus persicae* have indicated that exposure to lambda-cyhalothrin results in oxidative damage [60]. In Chinese honeybees, a serine protease is involved in defending against oxidative stress; notably, a study found that lambda-cyhalothrin treatment dramatically reduced the expression of this protease [61]. Future studies must be conducted to answer whether serine proteases are involved in oxidative stress protection in *Aedes aegypti*.

We also observed a marked decline in the regulation of cytochrome c oxidase subunits I, III, and NADH dehydrogenase subunits 2, 3, and 4L (Appendix A and Figure 6). Accordingly, enrichment scores of GO analysis explored the cytochrome c oxidase activity, NADH dehydrogenase activity, ATP-synthesis coupled electron transport, and respiratory chain terms (Figure 5B). ROS are produced in reactions in which the electron transport chain plays a primary role [62]. This downregulation of genes indicates shifts in enzymatic activities and metabolic processes that reduce energy production [49], resulting in less oxidative stress.

Cytochrome c oxidase subunit-III levels were detected as elevated in *Ae. aegypti* following permethrin exposure [63]. In the pupation of *Bactrocera dorsalis*, downregulation of NADH dehydrogenase and cytochrome c oxidase genes has also been identified [64].

Exploring the implications of electron chain transporter genes could account for the reduction in fitness observed in many resistant strains. Drops in respiration rate after exposure to pyrethroid insecticides, similar to those observed in the malaria vector *Anopheles coluzzi* [65], could explain this.

In previous work [7], our group has delved into the microbiota as a resistance mechanism in this Acacías strain and showed that removing its bacteriome produces a significant drop in resistance. We cannot rule out the possible involvement of microbiota in this strain’s resistance, especially its involvement in the high RR of AF13P. This novel understanding of the response mechanisms, including the role of enzymatic activity, opens up new avenues for further research and exploration.

Finally, we propose a modified model that incorporates novel mechanisms that complement the established insecticide resistance models based on *kdr* and metabolic resistance [10]. These mechanisms differ in their time of expression and the appearance of resistance factors in the insect, affirming the vast divergence of insecticide resistance mechanisms yet to be explored (Figure 7).

## 5. Conclusions

This study presents a novel perspective on the response of *Aedes aegypti* to lambda-cyhalothrin insecticide. Long-term exposure leads mosquitoes to express different proteins related to the cuticle, energetic metabolism, and protease synthesis. We propose a new model that incorporates these novel, yet complementary, mechanisms into the established insecticide resistance literature based largely on *kdr* and metabolic resistance. These data now provide the basis for expanding our understanding of complex mechanisms used by insects to resist insecticide treatments.

## Figures and Tables

**Figure 1 insects-16-00106-f001:**
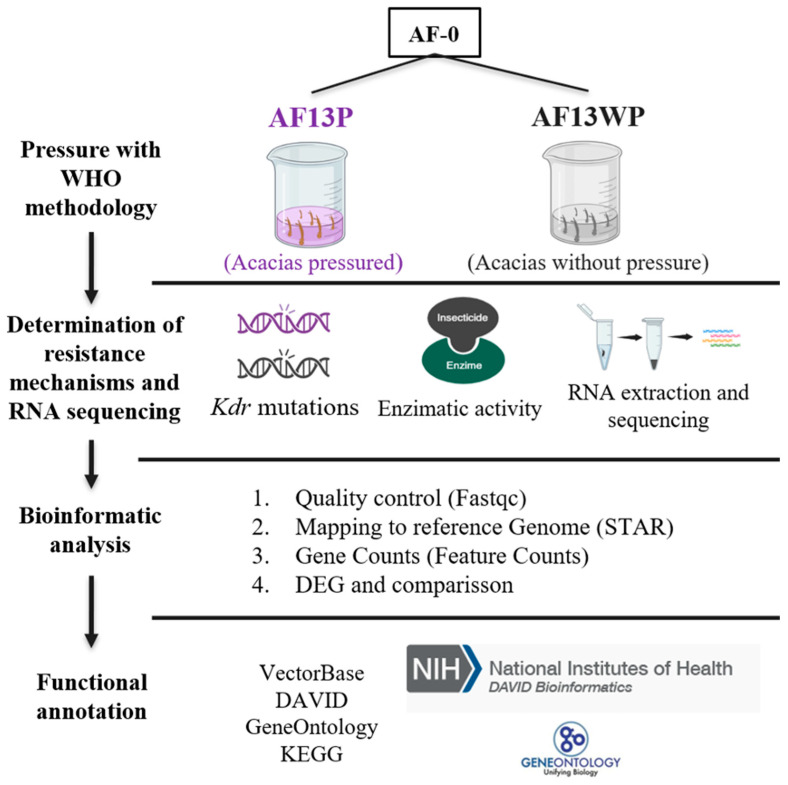
Experimental design overview. The Acacías population AF-0 was collected as stated by Granada et al. [16], and following WHO protocols, the pressure was performed at the larval stage. Three biological replicates were performed to extract RNA and proceed with sequencing. Vector Base, Gene Ontology categories, and DAVID were consulted to obtain functional annotation of the DEGs (Differentially Expressed Genes).

**Figure 2 insects-16-00106-f002:**
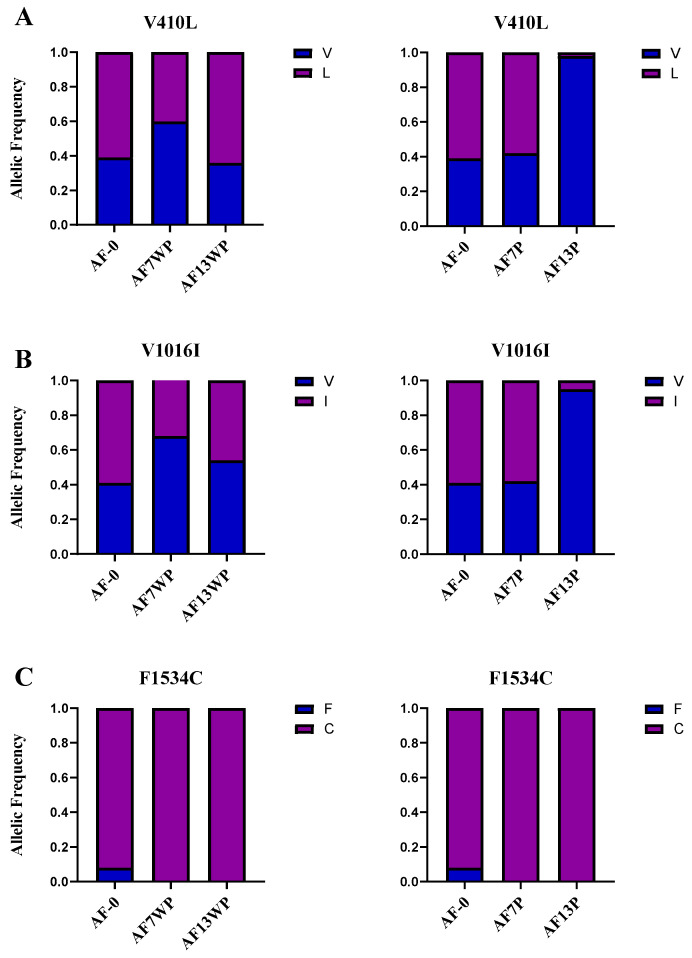
*Kdr* mutation frequencies for mutations (**A**) V410L, (**B**) V1016I, and (**C**) F1534C in unpressured (AF13WP) and pressured (AF13P) populations. Allelic frequencies of all the populations in this study. Blue represents susceptible alleles, and purple represents resistant alleles. The F7 pressured (AF7P) and unpressured (AF7WP) allelic frequencies data reported by Granada et al. [16] are also shown.

**Figure 3 insects-16-00106-f003:**
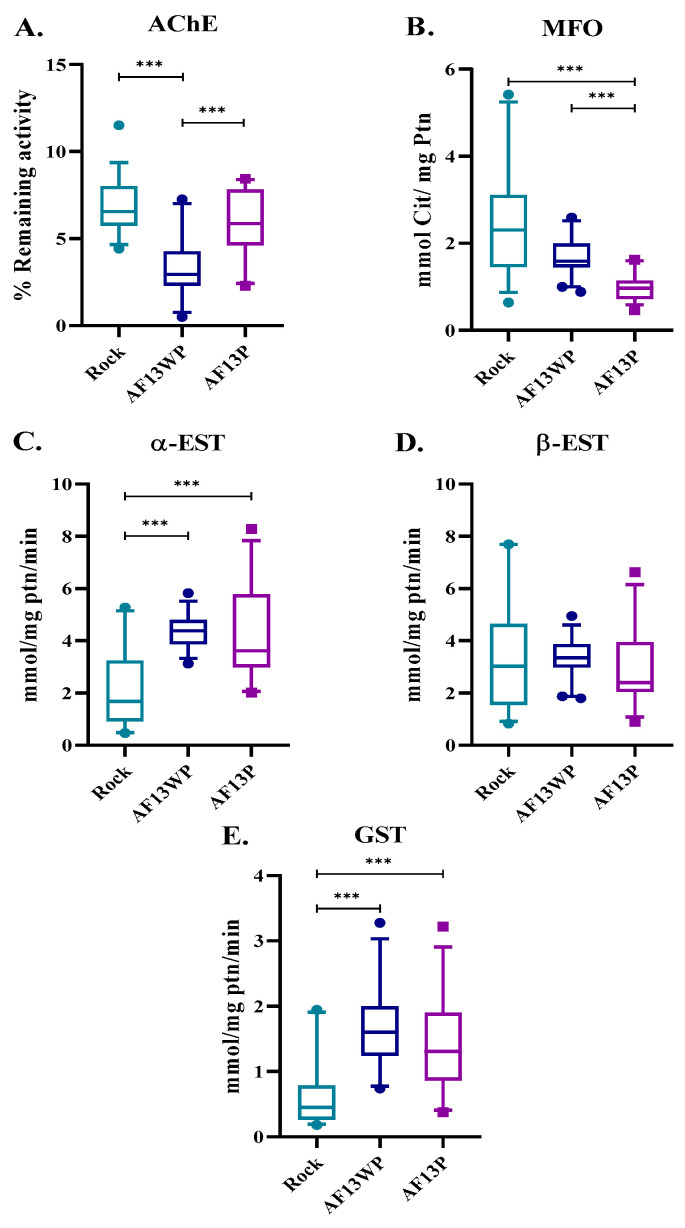
Enzyme activity for *Ae. aegypti* populations. (**A**) Acetylcholinesterase (AChE), (**B**) mixed function oxidase (MFO), (**C**) α-esterase (α-EST), (**D**) β-esterase (β-EST), and (**E**) Glutathione-S-transferase (GST). Forty mosquitoes were used for each assay. Boxes and whiskers correspond to mean and percentiles 5–95. Asterisks correspond to significant differences (Kruskal Wallis, *** = *p* < 0.0001).

**Figure 4 insects-16-00106-f004:**
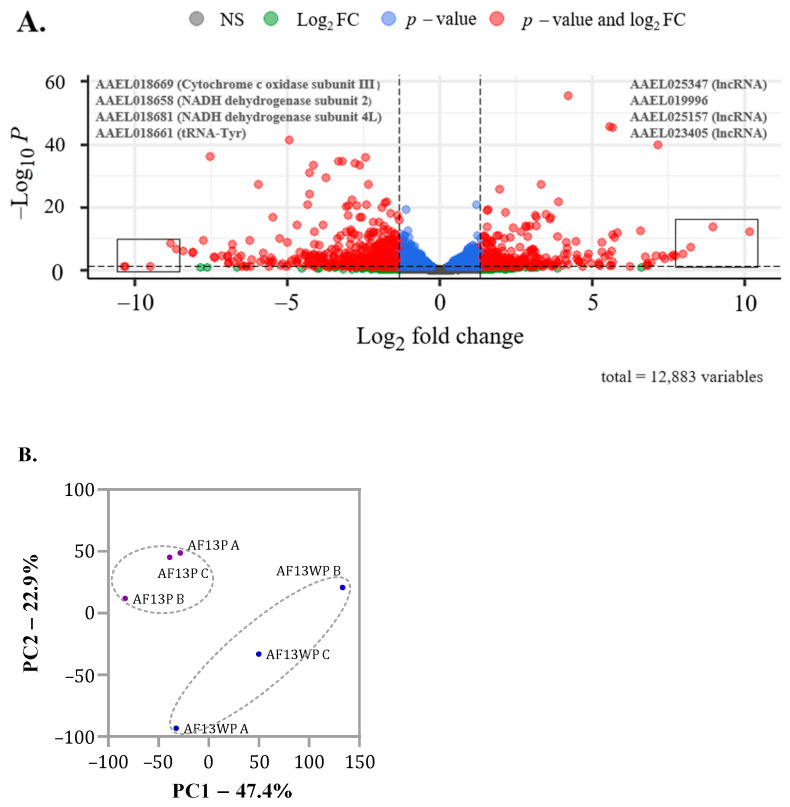
(**A**) Volcano Plot of False Discovery Rate vs. log 2-fold Change. The TOP 4 most downregulated and upregulated genes and their corresponding ID or annotation description are in the upper left and right corners. The same genes are in boxes in the lower left and right corners. Blue dots represent genes that surpassed only the FDR threshold of 0.05 but remained lower than log_2_FC of 1. Green dots are statistically non-significant (FDR > 0.05) but exceeded the log_2_FC threshold of 1. Meanwhile, the red dots correspond to statistically significant DEGs (FDR < 0.05) and have log_2_FC > 1. Grey dots have no statistical significance (NS) nor representative log_2_FC. FDR is represented as a negative log-transformed *p*-value (−Log_10_P). (**B**) PCA of intrasample variation. Each sample’s normalized and scaled counts were used to plot the Principal Component Analysis (PCA). Letters A, B, and C represent replicates for AF13P and AF13WP.

**Figure 5 insects-16-00106-f005:**
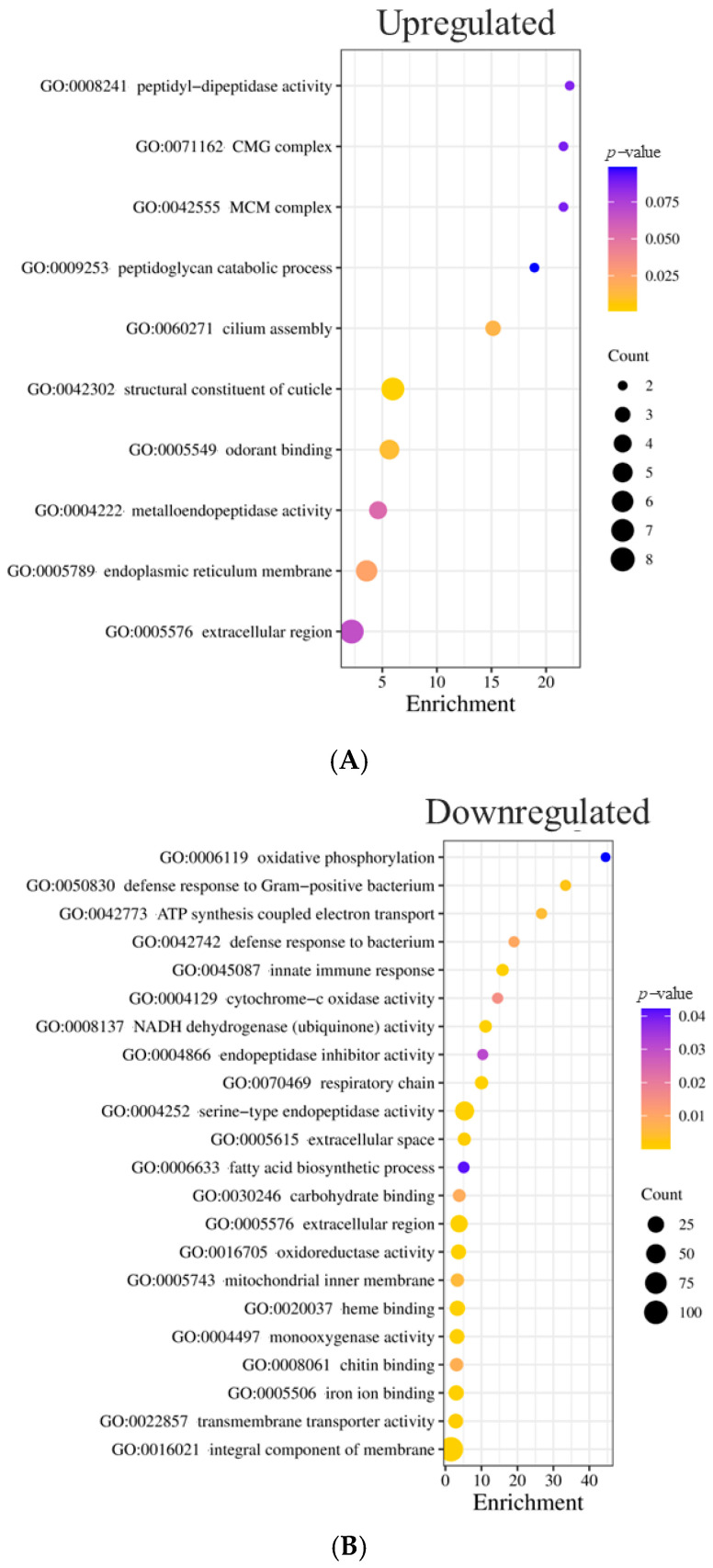
GO terms of upregulated (**A**) and downregulated (**B**) genes. The size of the bubbles corresponds to gene count in that specific term, while the color represents the associated *p*-value. Enrichment score corresponds to the number of genes with that term divided by the background.

**Figure 6 insects-16-00106-f006:**
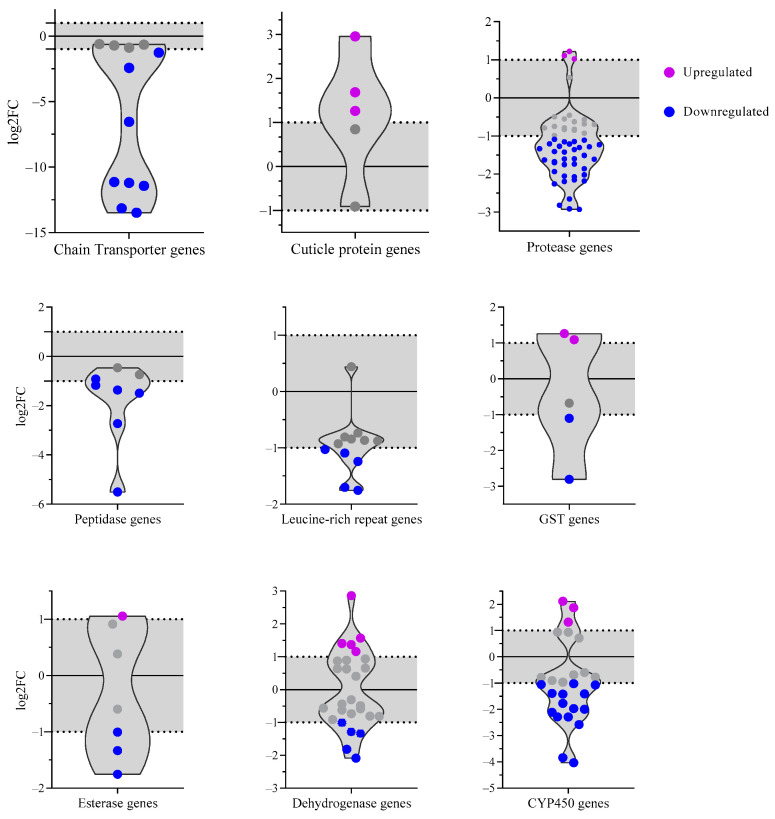
According to DAVID clustering and manual search, violin plots show specific gene regulation in 9 groups. The gray color corresponds to the area of no significant regulation. The dotted line determines the threshold of >1 and <−1 log_2_FC significant upregulated and downregulated genes. Blue dots represent specific downregulated genes, and purple dots indicate upregulated genes.

**Figure 7 insects-16-00106-f007:**
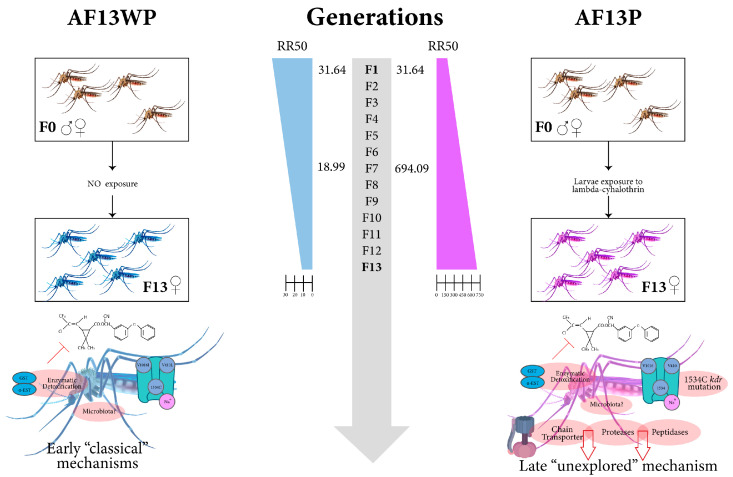
The model proposed for the response to Lambda-cyhalothrin insecticide in *Aedes aegypti*. Early and late mechanisms are indicated on the left and right sides. Classical mechanisms (“early”) correspond to *kdr* and metabolic resistance in response to insecticides. Unexplored mechanisms (“late”) correspond to mechanisms found in response to prolonged insecticide exposure in AF13P.

**Table 1 insects-16-00106-t001:** Resistance ratios to lambda-cyhalothrin in Acacías and the Rockefeller strains.

*Ae. aegypti* Population	Generation	Pressure	N	LC_50_ (95% CI)	LC_90_ (95% CI)	RR50	RR90	X^2^	df	Slope± SE
Rockefeller	N/A	N/A	1260	0.000474 (0.000398–0.000564)	0.0016 (0.0012–0.0022)	1	1	65	61	2.3± 0.1
Acacías Parental (AF-0) *	0	No	1260	0.015 (0.012–0.018)	0.05 (0.04–0.09)	31.64 **	31.25 **	265	52	2.1± 0.1
Acacías F7 WP *	7	No	1260	0.009 (0.007–0.010)	0.033 (0.026–0.044)	18.99 **	20.62 **	233	52	2.1± 0.1
Acacías F7 P *	7	Yes	1260	0.329 (0.252–0.430)	0.798 (0.644–1.080)	694.09 **	498.75 **	185	40	2.7± 0.2
Acacías F13 WP	13	No	840	0.005 (0.003–0.007)	0.017 (0.014–0.021)	10.54 **	10.62 **	21	40	2.4± 0.44
Acacías F13 P	13	Yes	840	0.355 (0.310–0.409)	0.767 (0.676–0.894)	748.94 **	479.38 **	68	40	3.1± 0.21

Bioassays are indicated with the corresponding LC_50_ and LC_90_ and their respective 95% confidence intervals (CIs). The Resistance Ratio (RR) was calculated by dividing the LC of the target population by that of the Rockefeller population [16]. * Data extracted from Granada et al. [16] ** Resistant populations, N: number of larvae used in the bioassay, N/A: not applicable; SE: standard error. Significant differences between LC_50_ and LC_90_ were assessed by the lethal dose ratios test [18]. All Acacías strains LC_50_ and LC_90_ presented significant statistical differences with a *p*-value < 0.05.

## Data Availability

All data presented in this article are available in five Excel files (see Appendix A). Moreover, analyses of transcriptome data are provided. The RNAseq datasets generated for this study are in the Gene Expression Omnibus (GEO) DataSets “https://www.ncbi.nlm.nih.gov/gds (accessed on 31 October 2024)” under the accession number GSE254270.

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
