# Peer review of "Long-Term Exposure to Lambda-Cyhalothrin Reveals Novel Genes Potentially Involved in Aedes aegypti Insecticide Resistance"

_insects, 2025, doi:10.3390/insects16020106_

Round 1
Reviewer 1 Report (Previous Reviewer 2)
Comments and Suggestions for Authors
Dear authors,
Thank you very much for the changes and adjustments in the manuscript.
Author Response
We are very grateful for your help and comments on our manuscript. Thanks.
Reviewer 2 Report (New Reviewer)
Comments and Suggestions for Authors
Please see attached document

Author Response
Reviewer’s report
Long term exposure to lambda-cyhalothrin reveals novel genes potentially involved in Aedes aegypti insecticide resistance
This article evaluated the effect of lambda-cyhalotrhin exposure for 13 generations on an insecticide resistant strain of Aedes aegypti from Columbia. The pressured strain was compared to the same generation (generation 13) as the unpressured stain as well as to an insecticide susceptible reference strain. The authors looked at several known kdr mutations know to confer resistance to pyrethroids in this strain, metabolic resistance mechanisms, and looked at differential gene expression between the pressured and non-pressured strain at generation 13. A model is proposed including the known target- site and metabolic mechanisms along with other mechanism suggested following their findings of differentially expressed genes.
Introduction
Line 51: use instead of used.. persists instead of persist.
R/ Thank you for your comment. Since we are talking about insecticides, we did not change this word.
Line 60: Also should not have a capital A
R/ Thank you for your comment. Corrected.
Materials and Methods:
Line 95: instead of ‘exposed to’.. I suggest ‘were offered’…
R/ Thank you for your comment. Corrected in line 95.
Line 95: were moist filter paper strips inserted into cages immediately after feeding or roughly after how long? Please specify
R/ Thank you for your comment. This was clarified in the new version (line 96).
Lines 101-102: please specify the starting size of each population if possible (rough size is fine)
R/Each population was started with at least 1000 individuals. We clarified this information in the new version (lines 106-107).
Line 113: Figure 1 legend Acacias spelling should be corrected- it says Acacia
R/ Thank you for your correction. Modified. Line 114.
Line 117: DEG should be written out in full as it has not yet been defined
R/ Thank you for your correction. We added the nomenclature’s meaning. Line 118.
Lines 127-137: more information on the statistical test is required- what statistical test does Lethal Dose Ration test use?
R/ We follow Robertson et al. to determine the statistical test by comparing overlapping confidence intervals (reference 18 in our manuscript). However, we agree with the reviewer and included further detail in lines 138-139. We also made this clarification in the legend of Table 1.
Line 140: Allelic should be allelic
R/ We agree. Thank you for the comment. Modified (line 142).
Lines 153-172: was the Rockefellar strain used as a susceptible reference strain? Please specify
R/ You are correct. We used Rockefeller as a reference strain. We specified further in line 157. Thanks to the reviewer.
Lines 167-168: please spell out DTNB and TMBZ at first use
R/ We agree. Thank you for the comment. We spelled out DTNB and TMBZ. Line 170 and 171.
Line 173: Bioinformatics should be bioinformatics
R/ Corrected. Line 177.
Line 190: has caps in all first letters of the section title. Please check your titles and make sure the use of caps/non-use of caps is consistent throughout your article
R/ We agree with the reviewer and changed our titles.
Results
Lines 225 and 229: AF0 is written differently please correct and use the same way throughout
R/ Thank you. We changed AF0 accordingly. Line 229 and 233.
Lines 231-232: Indicate the statistical test used
R/ As stated, the statistical test, called the Lethal Dose Ratios Test, involves overlapping 95 confidence intervals, as reported by Robertson et al. (reference 18).
Line 235: Table 1 heading – please include in the title that the comparisons are made between the Acacías strains and the Rockefeller insecticide susceptible strain.
R/ We accepted this recommendation and modified the Table 1 heading.
Line 259: the increased frequency of 1016I is given but does not show what it increased from- please include this
R/ We are very grateful to the reviewer. We clarified this section, adding each allelic frequency observed in Figure 2 (Lines 262 to 266 of the corrected document).
Line 260: similar to comment above
R/ We agree with the reviewer. We compare the changes between generations 7 and 13 of the pressured and unpressured populations.
Lines 266-267: please indicate where the AF7P and AF7WP data are from- was this part of this study or from the previous one?
R/ We thank the reviewer for the comment. We added the precedence of the data in line 257 and in the description of Fig 2 (line 272).
Line 281: by ‘susceptible counterpart’ do you mean AF13WP? Please rather specify this as AF13WP is not really susceptible, could perhaps say unpressured counterpart
R/ The reviewer is right. We corrected and specified AF13WP as stated in line 286.
Figure 3: please make the graphs larger, they could still fit side by side her and be quite a lot larger
R/ We agree. Graphs enlarged.
Line 288: all significance levels in the graphs appear to be *** p<0.0001, why include the key of all the other levels? Are the *s in the graphs correct- all being ***?
R/ The reviewer is correct. All comparations were ***. We have now only stated the significance of ***. Figure 3 description. Line 293
Line 289: in this heading rather say potential new insecticide resistance mechanisms
R/ We amended the recommendation. Thank you. Line 294.
Line 315: down and up regulated
R/ We changed it accordingly. Line 320.
Lines 323-324: it is interesting that there seems to be much more variability in AF13WP
R/ Indeed. However, we used the corrected p-value to search for genes; these discrepancies, especially in PCA1, did not affect our analysis.
Lines 388-389: Should be no full stop at end of title
R/ We modified this in the new version
Lines 340-383: it is not clear which genes are from the respiratory chain (as indicated in section title)- please highlight this in this section
R/ We agree with the reviewer. We clarified this in line 364.
Figure 5: please ensure the graphs text is similar in size for the up and down regulated gene graphs- the down-regulated one has been made much smaller
R/ We changed the size accordingly. Thank you for your comment.
Lines 401-402: speaking about metabolic genes in line 401 then say…. of these only the cuticle protein gene…. This makes it sound like the cuticle protein gene is a metabolic gene? Please reword this section for clarity
R/ We thank the reviewer for his comment. We rephrase it at line 407.
Line 432: please confirm if you have previously defined LRR- if not please define it
R/ We defined LRR (Leucine Rich Repeats). Thank you. Line 437.
Discussion
Line 445-446: instead of saying drastically diminished to 0.02 and 0.05 respectively rather give the % it was diminished by… It is not clear if the diminishment was drastic when there is no value to compare it to.
R/ We thank the reviewer for this insight. We have rephrased it to show the actual comparison between AF7P and AF13P (lines 450- 453).
Lines 449-450: instead of ‘with the pyrethroids increased resistance’ it should be with increased resistance to pyrethroids.
R/ We made the change at line 457.
Lind 459-462: what about lines 407-411? And Lines 238, 160, 143 in Table S1, and in the violin plot? Do you mean not observed in the biochemical tests?
R/ The reviewer is correct. We specified that no changes were observed in the metabolic enzymatic assays. Line 471.
Lines 469-472: it should perhaps also be mentioned that there are other kdr mutations which were not screened for in this study
R/ Thank you for the comment. We screen for the primary kdr mutations reported in Colombian Aedes aegypti strains. However, the reviewer is correct in pointing out other mutations. We clarified this in lines 460 – 462.
Line 478: do you mean overregulated or upregulated? Same comment for lines 481 and 487
R/ We corrected the word upregulated instead of overregulated for consistency, as the reviewer stated (lines 488, 497).
Line 489: do you mean after 4 or 5 generations have experienced insecticide selection pressure (i.e. each generation) or does it still happen if generation 1 only was pressured then you detect it after 4 or 5 generations after the pressure but without applying more pressure after generaton1? This needs to be written so this can be distinguished, similar for line 494
R/ Thank you to the reviewer for the comment. As the reviewer noted, this was addressed in lines 498–500 and 504–505. Continued pressure over several generations revealed genes that weren't upregulated after just one generation of selection but through continuous selection.
Lines 527-530: also thickened cuticle (measured using SEM) has been associated with an insecticide resistant strain of Anopheles funestus, not really a new mechanism
R/ The reviewer is correct. This is not surprising since insecticides have been found to be associated with cuticle reformation and cuticle thickening.
Line 536: …‘down-regulated in laboratory selected strains to Bt’… do you mean down-regulated in laboratory strains selected for resistance to Bt?
R/ No, we mean that this gene is downregulated in mosquito strains selected with Bacillus thuringiensis israelensis toxins. We Modified it in lines 546-547.
Line 544: by overregulated do you mean up and/or down regulated? If both, overregulated is appropriate
R/ Yes, overregulate means either upregulated or downregulated. We clarified this information in the latest version (554).
Line 553: please specify the insecticide
R/ We mentioned lambda-cyhalothrin in line 562, and immediately after, we mentioned the proteases of Periplaneta americana. With this, it is sufficient to indicate that the insecticide used is lambda-cyhalothrin.
Lines 565-566: please reword this sentence- which question?
R/ We agree with the reviewer. We re-phrased the sentence and included the question. Line 577.
Line 575: sub-unit 3 or III please confirm and change if needed
R/ The reviewer is right about the number. Changed at line 587.
Lines 581-587: It is not clear what comes first regarding point mutations in sodium channel gene, cuticle proteins, energetic metabolism, protease synthases etc. While differences were detected between the WP and P lines at F13 these were from parental line that already had some resistance to the insecticide tested. Please consider rewording.
Also it appears that transcriptomics was done at F13 (WP and P) only so how can one tell what the changes were over time- this would have required tests over more generations, e.g. F1, F7, F13… It is also too general a statement (584-587) as you have the data from this study- it can be proposed but more work would be needed to confirm the idea- make it more specific to what you found and less generalized
R/ The reviewer is correct. It would be interesting to evaluate this mosquito strain for more generations. However, we started this study knowing that strain AF-0 had kdr mutations and metabolic enzymes altered. This study was performed in 2016. We have followed this mosquito strain for generations, allowing us to propose different biological questions. One is related to changes in kdr mutations and identifying new insecticide resistance mechanisms.
Figure 7: AF13WP should not be called susceptible as it still had quite a high RR. Unless you mean the Rockefeller susceptible strain then please refer to it as such. What is the purpose of the F13 susceptible tile on the right hand side of Figure 7?
R/ The reviewer is correct. We changed the image accordingly. Thank you for the comment.
Line 603: ‘Early and late mechanisms’- is the timing of the appearance of these mechanisms actually known? Use caution.
R/ We are very grateful for the reviewer’s recommendation. Indeed, our data support the idea that due to low insecticide lambda-cyhalothrin concentrations, mosquitoes respond to insecticide exposure first by expressing metabolic enzymes and kdr mutations. We call this response classical or early. After prolonged exposure to this insecticide, these mechanisms are replaced by other unexplored mechanisms, which we call “late”. We presented this idea in Figure 7 and clarified this information in the figure legend.
Also line 605-606: the RR did not increase that much between AF7P and AF13P so how do you know these potential additional insecticide resistance mechanisms develop over a longer time?
R/ We agree with the reviewer. Indeed, this is an important question to study in the future. We know that kdr mutations and metabolic enzymes are not present in our higher resistant mosquitoes, but when the new mechanisms appear, it is another question to evaluate.
Supplementary tables:
S1: Rather use the strain names and not resistant and susceptible
S4: the tab labelled over-regulated should rather be labelled up-regulated
R/ Thank you for the recommendation. We modified the aforementioned supplementary tables.
Sincerely,
Omar Triana-Chavez
Corresponding author
This manuscript is a resubmission of an earlier submission. The following is a list of the peer review reports and author responses from that submission.
Round 1
Reviewer 1 Report
Comments and Suggestions for Authors
The present study is interesting and represents a relevance in the advancement of the study of the differential expression profile throughout generations of Ae. aegypti, independently of KDR mutations and metabolic resistance mechanisms, and how the interaction of different genes can support this result. I recommend its publication after taking into account the following observations.
Lines 46 and 48. The pages of references 1 and 2 are not available.
Line 140. Reference 17 is not available.
Methodology Line 121. Only the LC90 and LC50 were calculated?
Provide what software was used to estimate the LC50 and 90. Line 142. 10 ml of mosquito homogenate?
Line 196. Delete “but in the last six generations, the RR50 showed a slight increase (AF13P, RR50 748.94)”. The data are repetitive.
Results and discussion
Table 1. Please add a column with x2 and gl
What is the RR50 of the susceptible strain being compared against to obtain the value of 1?
It is inconsistent that, although the Rockefeller strain exactly matches the values ​​of the previous article referred to for the F7 generation, even in decimals and in the IC. Since these results of the susceptible strain were carried out previously, a recent evaluation of the susceptible strain should be carried out at the same time as the populations included in this work, that is, updating the LC's.
The above is mainly due to the difference shown by the enzymatic activity values ​​of the susceptible strain in terms of AChE, if it is the same strain and the same individuals there is a great inconsistency given that in Granada et al. 2021, this reference population presents a significantly different percentage than that reported in this article.
The same is true for MFO, the value is very different from what was previously reported, even 10 times higher, please expand the discussion on why this value decreases even in the population with selection. The same for alpha and beta-esterases.
Author Response
October 24, 2024
Dear Sirs
Insect Editorial Journal
Ref: Response to reviewers manuscript ID 3251384 “Long-term exposure to lambda-cyhalothrin reveals novel genes involved in Aedes aegypti insecticide resistance”.
First, the authors thank the reviewers and editor for their clear and concise comments. It is essential to state that most of the corrections were conducted according to reviewer recommendations.
We sincerely have adequately addressed all their concerns, and we are grateful to them for their time and attention.
Please find below the answers to the reviewers' evaluations.
Best Regards,
Omar Triana-Chavez
Corresponding author
Comments and Suggestions for Authors
Reviewer 1
The present study is interesting and represents a relevance in the advancement of the study of the differential expression profile throughout generations of Ae. aegypti, independently of KDR mutations and metabolic resistance mechanisms, and how the interaction of different genes can support this result. I recommend its publication after taking into account the following observations.
Lines 46 and 48. The pages of references 1 and 2 are not available.
R/ Reference 1 and 2 were checked. Its links are available.
Line 140. Reference 17 is not available.
R/ Reference 17 is available through its direct link.
Methodology Line 121. Only the LC90 and LC50 were calculated?
R/ The mortality assay was based on concentrations that ranged from 10% and 90%, and it permits the calculation of all possible LC of the corresponding mosquito populations to lambda-cyhalothrin. However, we used LC90 only for the pressure procedure. Moreover, to depict RR50 and RR90, we calculated LC50 and LC90. These information is clarified in the new version, section 2.3
Provide what software was used to estimate the LC50 and 90. Line 142. 10 ml of mosquito homogenate?
R/ The SPSS program was used to calculate the LC of mosquitoes. We added this information in (L128 in the revised manuscript).
R/ We are sorry for the mistake regarding the homogenate. We clarified this information in the new version. Mosquitoes were homogenized in 300 µl of deionized water, as stated in line 153, section 2.5.
Line 196. Delete “but in the last six generations, the RR50 showed a slight increase (AF13P, RR50 748.94)”. The data are repetitive.
R/ We accepted this recommendation.
Results and discussion
Table 1. Please add a column with x2 and gl
R/ We are very grateful for the reviewer’s recommendation. We added the Chi-squared and degrees of freedom to Table 1.
What is the RR50 of the susceptible strain being compared against to obtain the value of 1?
R/ The value of Rockefeller RR was divided by itself to obtain value 1.
It is inconsistent that, although the Rockefeller strain exactly matches the values ​​of the previous article referred to for the F7 generation, even in decimals and in the IC. Since these results of the susceptible strain were carried out previously, a recent evaluation of the susceptible strain should be carried out at the same time as the populations included in this work, that is, updating the LC's.
R/ Thank you for this suggestion. We evaluated the susceptibility of the strains AF13WP (13th generation without pressure) and AF13P (13th generation with pressure) simultaneously. Moreover, the Rockefeller susceptible strain is always used as a control in all the bioassay tests. This information is shown in Table 1. It is to be noted that these results are independent of the enzymatic activity assays.
The above is mainly due to the difference shown by the enzymatic activity values ​​of the susceptible strain in terms of AChE, if it is the same strain and the same individuals there is a great inconsistency given that in Granada et al. 2021, this reference population presents a significantly different percentage than that reported in this article.
R/ Thank you for the comment. The reviewer has a point. However, talking about enzymatic activity, our strains differ from the ones in Granada et al., 2021, since ours are 13 generations advanced (AF13P With and AF13WP without pressure) since the parental Acacias strain. It is also worth noting that we used a new mosquito batch for the Rockefeller enzymatic assays, which are displayed in Fig 3. Rockefeller mosquitoes are also used in each of the enzymes of AF13P and AF13WP, providing an internal control as required in protocol-reference 19.
The same is true for MFO, the value is very different from what was previously reported, even 10 times higher, please expand the discussion on why this value decreases even in the population with selection. The same for alpha and beta-esterases.
R/ We acknowledge some of the limitations of enzymatic assays. According to previous guidelines, enzymatic activity assays were executed using positive and negative controls. The Rockefeller mosquitoes were used as an internal control in all individual plates for each evaluated enzyme1. We corroborated enzymatic activity results by comparing our Rockefeller enzymatic levels with other studies on CYP4502,3, GST3, AChE, and Esterase4 enzymes. We concluded that they had similar concentrations, indirectly validating our enzymatic assays. Another aspect that should be acknowledged is that we illustrated enzymatic assays as a distribution of insect enzymatic activity. Individual levels within populations exhibiting elevated enzymatic activity may drive resistance shifts in wild populations. Our portrayal is suitable for comparing different populations rather than detecting these particular levels.
Reviewer 2 Report
Comments and Suggestions for Authors
Title: Long-term exposure to lambda-cyhalothrin reveals novel genes involved in Aedes aegypti insecticide resistance
The manuscript investigates the development of insecticide resistance in the mosquito species Aedes aegypti after long-term exposure to lambda-cyhalothrin. The study reveals that resistance mechanisms extend beyond classical genetic mutations and metabolic enzyme activity, involving changes in gene expression related to the cuticle, energy metabolism, and protease synthesis. The findings suggest a complex interplay of factors contributing to resistance, highlighting the need for comprehensive approaches in vector control strategies.
The manuscript is well written. However, some methods are missing detailed description, eg the method of the enzyme activity or the molecular methodologies. Furthermore, the logic and discussion about the upregulated enzymes is not very clear. As you found P450s upregulated why you did not consider them being the player for the RR? I suggest doing heterologous expressions with these candidates. I recommend this manuscript for publishing in insects after major revisions.
General comments:
- One of the main hypothesis is that no MFOs found upregulated in comparison to other pyrethroid-resistant mosquito strains. However, the method is not very well described and it is very blurry how you came to these conclusions. Therefore, it is crucial do describe the methods in detailed.
- I also recommend adding metabolic-microsomal studies.
- As you found P450s upregulated in the RNAseq why you haven’t conduct heterologous expression following metabolic analyses?
- These P450s you found upregulated might lead to the RR you are describing
- The PCR method needs more detailed description
- RNAseq analysis needs detailed description
- Some wordings can be changed
Detailed comments:
- Line 25: delete “is a problem”
- Line 30: “Our data supports a nuanced perspective in which the resistance of mosquitoes is likely influenced by additional mechanisms…” could be “Our data suggest that mosquito resistance is influenced by additional mechanisms…”
- Line 43: “impacting global health.”
- Line 45: “resulting in 40,000 deaths annually.”
- Line 57: explain kdr abbreviation before using it
- Line 62 + 63: Resistant to what insecticide? Maybe adding other KDR examples in Ae. aegypti mosquitoes?
- Line 87: “The larvae were raised in dechlorinated water and fed Purina® truchina fish food containing 48% protein” could be “Larvae were raised in dechlorinated water and fed Purina® truchina fish food (48% protein).”
- Line 115: adding the CAS number for Lambda cyhalothrin
- Line 121: dilution steps?
- Line 128: adding primer sequence and information, adding polymerase and other materials which were used, the method is not comprehensively written
- Line 139: how did you homogenize? Describe the whole procedure briefly
- Metabolic enzyme assay: to my understanding you are only measuring the content of proteins using the BCA assay? But where is method where you measured ENZYME ACTIVITY?
- Line 146: male or female mosquitoes were used? Explain briefly the extraction steps? Which concentration/yield of RNA did you receive?
- Line 152: I would rephrase “and low-quality sequences were cut (or “trimmed”) using Trimmomatic.” Also, since in the results some more information about which sequences were discarded, it would be good to explain the criteria of Trimmomatic step here as well.
- Line 154: “Gene counts were obtained through the Rsubread (DEseq2) package”
- Line 156: I would add the usual abbreviation “(log2FC)” in brackets, as it will be mentioned several times in the results.
- Line 156: I would add here which are the groups compared in the DE analysis, so just after FDR ≤ 0.05: “among the two compared groups AF13P and AF13WP”.
- Line 157-158: I would rephrase: “statistical analysis and further comparisons between the two groups“.
- Line 160: An explanation of the enrichment score mentioned many times in the results is missing.
- Line 161-163: whole paragraph not clear, difficult to read. Maybe a rephrase like: “The DAVID Gene system was used to elucidate resistance mechanisms by associating overrepresented genes in the RANseq data with specific biological processes reported in the Gene Ontology (GO) database and by performing pathway enrichment analysis using KEGG.”
- Line 165-169: not clear, getting a gene from where? “it is a modified Fisher exact P-value to test the probability of getting a gene (or a set of genes), whose term in the dataset is obtained with a particular frequency, over the total gene set of a background genome (in this case, Aedes aegypti AegL5 genome),
- Line 169: What? “The examination of the highest overall EASE scores within a group term, conveying a similar biological significance, was undertaken.”
- Line 174-175: “The GO annotations…”
- Line 177: “… and Biological Processes (BP) categories.”
- Line 178: “… and GO analysis.”
- Line 187: “We explored the effect of insecticide pressure on the R.R. of AF13P and compared it to population AF13WP using bioassays” could be “We explored the effect of insecticide pressure on the resistance ratio (R.R.) of AF13P compared to AF13WP using bioassays.”
- Line 229: Please add the respective method to the M&M section. The results can not be interpreted when the according method is missing
- Line 254: “insecticide-refractory” sounds not good to me.
- Line 256-257: since the quality control processes are included in the trimming, could be rephrased: “we obtained 790,369,834 total trimmed reads, with 93% mapping coverage on the reference AegL5 genome”. Also, is it exactly 93%? Looks more like “about” 93%.
- Line 259: “This genome”, which genome? I suppose Ae. aegypti, in this case why to give here in the results information about numbers of genes and coding proteins?
- Line 261-262: “of which 78.79% were uniquely mapped genes and 13.64% were 261 mapped to multiple loci“ – the meaning is that these two percentages refer to the 12,893 genes remaining, but the sum of the two percentages is not 92,4%... is the sum referring to the mapping coverage percentage, which is reported as 93%?
- Line 263: “(DEGs)”. In general, all the next repetitions should be “DEGs” and not “DEG”.
- Line 265: it is needed to be specified that 302 genes are upregulated and 602 downregulated compared to the AF13WP susceptible/unpressured population.
- Line 266: “DEGs”
- Line 267-268: “… and AF13WP populations for the Principal Component 1 (PC1) and 2 (PC2), contributing to…”
- Line 271: I would rephrase the last two points as: “… protein-coding genes. Among them, 230 were successfully annotated, while 245 were reported as hypothetical genes”.
- Line 276: I would separate the caption in (A) and (B) in order to divide the text between the two figures, since there are no common points discussed in the caption.
- Line 277: “up- and downregulated DEGs”. Rephrasing needed for the whole sentence, something like this: “For the TOP 4 highest up- and downregulated genes, accession numbers and gene descriptions are reported in the right and left side of the box, respectively”. The word box could be changed as well with “graph”, I don’t know if it was chosen for a specific reason.
- Line 278: the whole description of the dots is muddy. What is the p-value threshold for blue dots, 0.05? “Green dots passed only log2FC”, this has no meaning; actually, from the Figure 4 caption, green dots look to have log2FC≤1.
- Line 290: why is AAEL018671 missing from the Volcano Plot figure? Also, in the figure the TOP 4 downregulated genes are reported, while here in the text 6 genes are mentioned. Are they the TOP 6? It is not clear the reason of the choice of AAEL018669 and AAEL01866: was it only because of their annotation probably resistance-related?
- Line 298: “in the resistant strain”.
- Line 298-299: “we studied the biological role of up- and downregulated genes and …”
- Line 302: maybe “analyze” better than “describe”.
- Line 303-304: “Using DAVID on the list of 320 up-regulated genes within the broader context of the entire dataset, nine distinct clusters were reveal”.
- Line 305-307: “The first gene cluster, characterized by an enrichment score of 4.3%, includes conserved… and a pathogenesis-stress-related protein…”
- Line 308: “The second cluster (enrichment score of 2.5%) comprises…”
- Line 309-310: “Cytochrome P450, although having an enrichment score <1%, emerged from the data as a third cluster”. This needs to be explained: what does “emerges among these data” means? Beside of the biological importance of P450s in resistance, this finding needs to be justified better.
- Line 311: “overexpressed” better than “overregulated”.
- Line 312: “p-values are structural constituents… and odorant…” Why is “p-value” here italic while elsewhere it is normally written?
- Line 313: portraying a similar picture… how? Structural constituents of the cuticle is right, but odorant binding was not mentioned before…
- Line 315: “included six statistically significant clusters (enrichment score above 1)”.
- Line 317: “comprised”.
- Line 319: I would switch the . with a ; and then: “the terms GO:0005506 ion binding and GO:0004129 cytochrome-c oxidase activity were also significant”.
- Line 320: “The fourth and fifth cluster included terms related to…”; I would also delete the “(Figure 5)” since all the paragraph is shown in the figure, which may be kept just mentioned at line 322.
- Line 321: “such as GO:002857…”
- Line 322: “(Figure 5B)”, not generally Figure 5.
- Line 322: “The last cluster showing an enrichment score above 1 incorporated leucine-riche repeats, but it was not represented by GO terms”.
- Line 326: I think that the writings in the figure should have the same character size.
- Line 327: “GO terms of up-regulated (A) and down-regulated (B) genes. The size of the bubbles corresponds to gene count in that specific term, while the color represents the associated p-value”.
- Line 329: The meaning of enrichment score was not explained before, it would be a good idea to explain it in the Materials and Methods.
- Line 333-335: A rephrasing would be better: “In terms of insecticide resistance-related genes, the most interesting categories for the reported DEGs were cuticle protein genes for up-regulated genes and chain transporter and stress-related proteins (proteases, peptidases, leucine-rich repeat genes) for the down-regulated genes”.
- Line 335: “We assessed four categories” in what sense? What is the conclusion about these four categories?
- Line 397: “Conventionally, mosquitoes exhibit primary resistance mechanisms characterized by alterations in the VGSC gene and metabolic enzymes” could be “Mosquitoes typically exhibit primary resistance mechanisms through VGSC gene alterations and metabolic enzymes.”
- Line 414: you have shown that CYP9J29 and CYP6AG4 are upregulated? How you conducted the other MFO assays?
- 4.3 if you discuss cuticular resistance mechanisms please insert them also in the introduction
- Line 429: “Our data supports a nuanced perspective in which the resistance of AF13P mosquitoes is likely influenced by additional mechanisms that are difficult to address using only classical mechanisms” could be “Our data suggest that AF13P mosquito resistance is influenced by additional mechanisms beyond classical ones.”
- Line 563: “After long-term exposure, mosquitoes respond to insecticides by expressing different proteins involved in the cuticle, energetic metabolism, and synthesis of proteases” could be “Long-term exposure leads mosquitoes to express different proteins related to the cuticle, energy metabolism, and protease synthesis.”
Comments on the Quality of English LanguageThe quality of english is acceptable with minor changes to some phrases (also stated in the above section)
Author Response
Title: Long-term exposure to lambda-cyhalothrin reveals novel genes involved in Aedes aegypti insecticide resistance
The manuscript investigates the development of insecticide resistance in the mosquito species Aedes aegypti after long-term exposure to lambda-cyhalothrin. The study reveals that resistance mechanisms extend beyond classical genetic mutations and metabolic enzyme activity, involving changes in gene expression related to the cuticle, energy metabolism, and protease synthesis. The findings suggest a complex interplay of factors contributing to resistance, highlighting the need for comprehensive approaches in vector control strategies.
The manuscript is well written. However, some methods are missing detailed description, eg the method of the enzyme activity or the molecular methodologies. Furthermore, the logic and discussion about the upregulated enzymes is not very clear. As you found P450s upregulated why you did not consider them being the player for the RR? I suggest doing heterologous expressions with these candidates. I recommend this manuscript for publishing in insects after major revisions.
General comments:
- One of the main hypothesis is that no MFOs found upregulated in comparison to other pyrethroid-resistant mosquito strains. However, the method is not very well described and it is very blurry how you came to these conclusions. Therefore, it is crucial do describe the methods in detailed.
R/ We thank the reviewer for this comment. We amended our methods and detailed qPCR reactions, biochemical studies, RNAseq, and DAVID clustering.
I also recommend adding metabolic-microsomal studies.
R/ Thank you for your insight. As CYP450 and esterases are established mainly in the membrane of the endoplasmic reticulum and the fragmentation of the mosquito produces microsomes, we believe that the literature we have added is enough to depict an orientation to these groups of enzymes.
- As you found P450s upregulated in the RNAseq why you haven’t conduct heterologous expression following metabolic analyses?
R/ Thank you for your comment. Although heterologous expression has been done in the past, it’s a time- and labor-intensive procedure that requires refining several factors1. It is out of the scope of our research. Nonetheless, it is a growing field that must be encompassed in future studies and will elucidate metabolic resistance mechanisms. Additionally, we found CYP450 overexpressed, which has been found in other studies on Aedes aegypti2. This field is excellent for subsequent studies and is considered inside our group.
- These P450s you found upregulated might lead to the RR you are describing
R/ We agree with the reviewer. However, as we found three upregulated CYP450 genes and no response in the enzymatic assays, we established that some other mechanisms are at play in this strain.
- The PCR method needs more detailed description
R/ Thank you for your comment. Further detailed method was added. Line 137, section 2.4. However, we would like to highlight that these procedures have already been published. We added the corresponding references.
- RNAseq analysis needs detailed description
R/ Thank you for your comment. Further detailed method was added. RNA sequencing, line 173, section 2.6.
- Some wordings can be changed
Detailed comments:
- Line 25: delete “is a problem”
R/ We thank the reviewer. We deleted it. Line 28.
- Line 30: “Our data supports a nuanced perspective in which the resistance of mosquitoes is likely influenced by additional mechanisms…” could be “Our data suggest that mosquito resistance is influenced by additional mechanisms…”
R/ We rephrase it to be more concise. Line 34.
- Line 43: “impacting global health.”
R/ We agree with the reviewer. Change done. Line 48.
- Line 45: “resulting in 40,000 deaths annually.”
R/ We agree with the reviewer. Change done. Line 49.
- Line 57: explain kdr abbreviation before using it
R/ Thank you for the appreciation. We added the abbreviation.
- Line 62 + 63: Resistant to what insecticide? Maybe adding other KDR examples in Ae. aegypti mosquitoes?
R/ We agree with the reviewer. We added more examples depicting our tested kdr mutations. Line 66-68.
- Line 87: “The larvae were raised in dechlorinated water and fed Purina® truchina fish food containing 48% protein” could be “Larvae were raised in dechlorinated water and fed Purina® truchina fish food (48% protein).”
R/ Modified according to your suggestion. Line 93.
- Line 115: adding the CAS number for Lambda cyhalothrin
R/ Thank you. CAS number added.
- Line 121: dilution steps?
R/ We clarified the concentrations that killed 10 to 90% of insects.
- Line 128: adding primer sequence and information, adding polymerase and other materials which were used, the method is not comprehensively written
R/ Thank you for the suggestion. We used the qPCR methods stated in Pareja-Loaiza et al3 . For improved readability; any reader could refer to the method section to delve into it. Line 144.
- Line 139: how did you homogenize? Describe the whole procedure briefly
R/ Thank you for the suggestion. We added more information on the homogenization of samples. Line 139.
- Metabolic enzyme assay: to my understanding you are only measuring the content of proteins using the BCA assay? But where is method where you measured ENZYME ACTIVITY?
R/ We agree with the reviewer. We added information and details regarding enzymatic activity bioassays that is depicted in reference 19. Line 164, section 2.5.
Enzymatic activity assays and kdr mutations are well described in the references as mentioned earlier in the manuscript. However, we described each one in more detail.
- Line 146: male or female mosquitoes were used? Explain briefly the extraction steps? Which concentration/yield of RNA did you receive?
R/ Thank you for the comment. We added information on RNA extraction. Only females were used for the whole procedure. Line 169, section 2.6.
- Line 152: I would rephrase “and low-quality sequences were cut (or “trimmed”) using Trimmomatic.” Also, since in the results some more information about which sequences were discarded, it would be good to explain the criteria of Trimmomatic step here as well.
R/ Thank you for your appreciation. We used the standard criteria of trimommatic to remove adapters, and trim phred scored below 30. We rephrased it to depict that. Lines 174-175.
- Line 154: “Gene counts were obtained through the Rsubread (DEseq2) package”
R/ Thank you for your comment. Word changed. Line 178.
- Line 156: I would add the usual abbreviation “(log2FC)” in brackets, as it will be mentioned several times in the results.
R/ Thank you for your comment. Abbreviation added. Line 180.
- Line 156: I would add here which are the groups compared in the DE analysis, so just after FDR ≤ 0.05: “among the two compared groups AF13P and AF13WP”.
R/ Thank you for the clarification. We rephrase it. Line 181.
- Line 157-158: I would rephrase: “statistical analysis and further comparisons between the two groups“.
R/ Thank you for the recommendation. Phrase added. Line 182.
- Line 160: An explanation of the enrichment score mentioned many times in the results is missing.
- We agree with the reviewer and included a new paragraph in line 199, section 2.7.
- Line 161-163: whole paragraph not clear, difficult to read. Maybe a rephrase like: “The DAVID Gene system was used to elucidate resistance mechanisms by associating overrepresented genes in the RANseq data with specific biological processes reported in the Gene Ontology (GO) database and by performing pathway enrichment analysis using KEGG.”
R/ We are very grateful for the reviewer’s recommendation. The sentence was modified at line 185, section 2.7.
- Line 165-169: not clear, getting a gene from where? “it is a modified Fisher exact P-value to test the probability of getting a gene (or a set of genes), whose term in the dataset is obtained with a particular frequency, over the total gene set of a background genome (in this case, Aedes aegypti AegL5 genome),
R/ We agree with the reviewer; the sentence was modified accordingly in lines 189 to 193 for clarity.
- Line 169: What? “The examination of the highest overall EASE scores within a group term, conveying a similar biological significance, was undertaken.”
R/ We removed this sentence and added a summary paragraph at lines 199-203.
- Line 174-175: “The GO annotations…”
R/ We accept the reviewer’s recommendation.
- Line 177: “… and Biological Processes (BP) categories.”
R/ We accept the reviewer’s recommendation.
- Line 178: “… and GO analysis.”
R/ We amended the clarifications made by the reviewer from line 200 to line 208.
- Line 187: “We explored the effect of insecticide pressure on the R.R. of AF13P and compared it to population AF13WP using bioassays” could be “We explored the effect of insecticide pressure on the resistance ratio (R.R.) of AF13P compared to AF13WP using bioassays.”
R/The abbreviation of RR is already in experimental design. Corrected phrasing. Line 216.
- Line 229: Please add the respective method to the M&M section. The results can not be interpreted when the according method is missing
R/ We specified the method section of enzymatic activity assays. We thank the reviewer for this recommendation.
- Line 254: “insecticide-refractory” sounds not good to me.
R/ We agree with the reviewer. We changed to “insecticide resistance phenotype”, line 283
- Line 256-257: since the quality control processes are included in the trimming, could be rephrased: “we obtained 790,369,834 total trimmed reads, with 93% mapping coverage on the reference AegL5 genome”. Also, is it exactly 93%? Looks more like “about” 93%.
R/ The reviewer is correct. It is approximately 93%.
- Line 259: “This genome”, which genome? I suppose Ae. aegypti, in this case why to give here in the results information about numbers of genes and coding proteins?
The reviewer has a point. This information is now in RNA seq methods.
- Line 261-262: “of which 78.79% were uniquely mapped genes and 13.64% were 261 mapped to multiple loci“ – the meaning is that these two percentages refer to the 12,893 genes remaining, but the sum of the two percentages is not 92,4%... is the sum referring to the mapping coverage percentage, which is reported as 93%?
R/ We agree with the reviewer. This information is now in lines 284-288 in context with the percentage of mapped reads and supplementary Table 1.
- Line 263: “(DEGs)”. In general, all the next repetitions should be “DEGs” and not “DEG”.
R/ It was modified in all the text.
- Line 265: it is needed to be specified that 302 genes are upregulated and 602 downregulated compared to the AF13WP susceptible/unpressured population.
R/ Thank you. Comparation clarified. Line 294.
- Line 266: “DEGs”
R/ We agree. Now and onward DEG have been changed to DEGs
- Line 267-268: “… and AF13WP populations for the Principal Component 1 (PC1) and 2 (PC2), contributing to…”
R/ Modified. Line 295.
- Line 271: I would rephrase the last two points as: “… protein-coding genes. Among them, 230 were successfully annotated, while 245 were reported as hypothetical genes”.
R/ The sentence was modified accordingly at lines 299.
- Line 276: I would separate the caption in (A) and (B) in order to divide the text between the two figures, since there are no common points discussed in the caption.
R/ The reviewer is right. We separated the caption.
- Line 277: “up- and downregulated DEGs”. Rephrasing needed for the whole sentence, something like this: “For the TOP 4 highest up- and downregulated genes, accession numbers and gene descriptions are reported in the right and left side of the box, respectively”. The word box could be changed as well with “graph”, I don’t know if it was chosen for a specific reason.
R/ We chose a box to depict the genes mentioned in the upper left and right corner. Thank you for your appreciation. We rephrased it in line 305.
- Line 278: the whole description of the dots is muddy. What is the p-value threshold for blue dots, 0.05? “Green dots passed only log2FC”, this has no meaning; actually, from the Figure 4 caption, green dots look to have log2FC≤1.
R/ Thank you for the appreciation. We have rephrased the whole description of the dots in the volcano plot. Line 308.
- Line 290: why is AAEL018671 missing from the Volcano Plot figure? Also, in the figure the TOP 4 downregulated genes are reported, while here in the text 6 genes are mentioned. Are they the TOP 6? It is not clear the reason of the choice of AAEL018669 and AAEL01866: was it only because of their annotation probably resistance-related?
R/ The reviewer is right. We corrected the number of the gene in the text (AAEL018661 is the correct one). Those additional two genes (AAEL018669 and AAEL01866) are depicted in Supplementary Table 1 and are shown in Fig. 6 DAVID clustering as chain transporter genes. Nonetheless, they provide no information in this context. We changed it in line 335 to depict only genes shown in the volcano plot.
- Line 298: “in the resistant strain”.
R/ We thank the reviewer for this comment. We accepted the recommendation.
- Line 298-299: “we studied the biological role of up- and downregulated genes and …”
R/ Thank you. We agree with the reviewer and amended our sentence at line 328.
- Line 302: maybe “analyze” better than “describe”.
R/ We agree with the reviewer. Word changed. Line 331.
- Line 303-304: “Using DAVID on the list of 320 up-regulated genes within the broader context of the entire dataset, nine distinct clusters were reveal”.
R/ We agree with the reviewer. Sentence changed. Line 332.
- Line 305-307: “The first gene cluster, characterized by an enrichment score of 4.3%, includes conserved… and a pathogenesis-stress-related protein…”
R/ We thank the reviewer for all the clarification annotations. Sentence changed. Line 334.
- Line 308: “The second cluster (enrichment score of 2.5%) comprises…”
R/ Sentence changed accordingly.
- Line 309-310: “Cytochrome P450, although having an enrichment score <1%, emerged from the data as a third cluster”. This needs to be explained: what does “emerges among these data” means? Beside of the biological importance of P450s in resistance, this finding needs to be justified better.
R/ Besides its relation with resistance, these genes in the third cluster are enriched according to DAVID, as depicted in Supplementary Table S2. That’s the reason behind “emerges among these data”.
- Line 311: “overexpressed” better than “overregulated”.
R/ Modified accordingly. Line 340.
- Line 312: “p-values are structural constituents… and odorant…” Why is “p-value” here italic while elsewhere it is normally written?
R/ We homogenized this word in all the text.
- Line 313: portraying a similar picture… how? Structural constituents of the cuticle is right, but odorant binding was not mentioned before…
R/ We refer to the coincidence between Gene Ontology and DAVID, which display the terms associated with cuticle and odorant binding genes. We made clear that difference. Line 342.
- Line 315: “included six statistically significant clusters (enrichment score above 1)”.
R/ Thank you. Modified. Line 344.
- Line 317: “comprised”.
R/ Thank you. Modified. Line 362.
- Line 319: I would switch the . with a ; and then: “the terms GO:0005506 ion binding and GO:0004129 cytochrome-c oxidase activity were also significant”.
R/ Agreed. Line 347.
- Line 320: “The fourth and fifth cluster included terms related to…”; I would also delete the “(Figure 5)” since all the paragraph is shown in the figure, which may be kept just mentioned at line 322.
R/ Modified. Line 349.
- Line 321: “such as GO:002857…”
R/ Modified. Line 351.
- Line 322: “(Figure 5B)”, not generally Figure 5.
R/ Modified. Line 351.
- Line 322: “The last cluster showing an enrichment score above 1 incorporated leucine-riche repeats, but it was not represented by GO terms”.
R/ Reviewer is right. Corrected the tense of the sentence.
- Line 326: I think that the writings in the figure should have the same character size.
R/ Because of the space limitations of the amount of GO terms displayed in Fig 5B compared to Fig 5A, they appeared as different character size.
- Line 327: “GO terms of up-regulated (A) and down-regulated (B) genes. The size of the bubbles corresponds to gene count in that specific term, while the color represents the associated p-value”.
R/ Thank you. We modified the connectors. Line 356.
- Line 329: The meaning of enrichment score was not explained before, it would be a good idea to explain it in the Materials and Methods.
R/ We thank the reviewer for this detail. As mentioned earlier, the meaning of the enrichment score was detailed in methods.
- Line 333-335: A rephrasing would be better: “In terms of insecticide resistance-related genes, the most interesting categories for the reported DEGs were cuticle protein genes for up-regulated genes and chain transporter and stress-related proteins (proteases, peptidases, leucine-rich repeat genes) for the down-regulated genes”.
R/ We thank the reviewer for the rephrasing. The paragraph changed at line 363.
- Line 335: “We assessed four categories” in what sense? What is the conclusion about these four categories?
R/ We provide this paragraph in anticipation of the upcoming discussion on the distribution of DEGs in each category. In addition to the cuticle protein genes, chain transporters, and stress-related proteins, we have expanded our focus to include four additional gene categories (GST, esterase, dehydrogenase, and CYP450). These modifications are in line 369. This expanded list provides a more comprehensive framework for analyzing the DEGs.
- Line 397: “Conventionally, mosquitoes exhibit primary resistance mechanisms characterized by alterations in the VGSC gene and metabolic enzymes” could be “Mosquitoes typically exhibit primary resistance mechanisms through VGSC gene alterations and metabolic enzymes.”
R/ Thank you for the revision. We perform the change in line 431.
- Line 414: you have shown that CYP9J29 and CYP6AG4 are upregulated? How you conducted the other MFO assays?
R/ They are upregulated in our data set, as shown in Supplementary Table 1 (DEG tab) and Fig 6 (CYP450 category). As requested, enzymatic assays for CYP450 are portrayed in the methodology of enzymatic activity assays. MFO assays detect the overall activity of all CYP450 enzymes. Those in the endoplasmic reticulum's membrane are transformed into microsomal structures through maceration and then detected.
- 4.3 if you discuss cuticular resistance mechanisms please insert them also in the introduction
R/ We appreciate the comment. We approach cuticle resistance as a finding that deviates from the “classical” approaches since it is uncommon for public health authorities to search for this mechanism. Due to this, we prefer to maintain the introduction highlighting only the classical approaches, kdr mutations, and metabolic enzymes, and then discuss the involvement of these other mechanisms in late response to insecticides.
- Line 429: “Our data supports a nuanced perspective in which the resistance of AF13P mosquitoes is likely influenced by additional mechanisms that are difficult to address using only classical mechanisms” could be “Our data suggest that AF13P mosquito resistance is influenced by additional mechanisms beyond classical ones.”
R/ We thank the reviewer for the comment. We changed this paragraph at line 460.
- Line 563: “After long-term exposure, mosquitoes respond to insecticides by expressing different proteins involved in the cuticle, energetic metabolism, and synthesis of proteases” could be “Long-term exposure leads mosquitoes to express different proteins related to the cuticle, energy metabolism, and protease synthesis.”
R/ We thank the reviewer for all the comments and helpful insights in this revision. We modified this paragraph at lines 597-598.
Round 2
Reviewer 2 Report
Comments and Suggestions for Authors
Thank you for taking the recommendations into consideration.